# Influence of Fine Grains on the Bending Fatigue Behavior of Two Implant Titanium Alloys

**DOI:** 10.3390/ma14010171

**Published:** 2020-12-31

**Authors:** Xiaojian Cao, Jiangpei Zhu, Fei Gao, Zhu Gao

**Affiliations:** 1School of Transportation & Civil Engineering, Nantong University, Nantong 226019, China; cxj1983@ntu.edu.cn (X.C.); zhujiangpei2957@126.com (J.Z.); 2Department of Technology, Shangdong Huawin Hauck-Energy Technology Co., Ltd., Jinan 250101, China; feigao_hauckenergy@126.com

**Keywords:** nanocrystal, surface modification, titanium alloy, fatigue

## Abstract

By means of the ultrasonic surface impact (amplitude of 30 μm, strike number of 48,000 times/mm^2^), nanograins have been achieved in the surfaces of both Ti6Al4V(TC4) and Ti3Zr2Sn3Mo25Nb(TLM) titanium alloys, mainly because of the dislocation motion. Many mechanical properties are improved, such as hardness, residual stress, and roughness. The rotating–bending fatigue limits of TC4 and TLM subjected to ultrasonic impact are improved by 13.1% and 23.7%, separately. Because of the bending fatigue behavior, which is sensitive to the surface condition, cracks usually initiate from the surface defects under high stress amplitude. By means of an ultrasonic impact tip with the size of 8 mm, most of the inner cracks present at the zone with a depth range of 100~250 μm in the high life region. The inner crack core to TC4 usually appears as a deformed long and narrow α-phase, while the cracks in TLM specimens prefer to initiate at the triple grain boundary junctions. This zone crosses the grain refined layer and the deformed coarse grain layer. With the gradient change of elastic parameters, the model shows an increase of normal stress at this zone. Combined with the loss of plasticity and toughness, it is easy to understand these fatigue behaviors.

## 1. Introduction

It is well known that stainless steels and titanium alloys are the main materials for manufacturing medical instruments and biomedical implants. In view of the high strength-to-weight ratio and the excellent corrosion resistance, implants are more likely to be made of titanium alloys. Since titanium alloy was chosen to make dental implants firstly, it has gone through three stages in the past 70 years. Pure Ti, Ti-Al base series, and the third-generation titanium alloys appear successively. Ti6Al4V (TC4) is one of the widely-used Ti-Al base alloys for artificial bones. According to the long-term clinical feedback, the in vitro biocompatibilities and the mechanical compatibilities have been the research focuses. From the perspective of easy processing and the follow-up treatment, ZrO gradually replaced the status of pure Ti in dental implants. The element aluminum is found as the reason of dementia, cerebral injury, and anemia. In addition, vanadium is harmful to the digestive system and the nerve center of the human body. Besides, prosthetic loosening, osteonecrosis, and bone degeneration usually occur because of the stress shielding. This stress shielding is caused by the large difference between implants and bones. The ultimate aim of medical implants is to improve the in vivo biological safety and the persistent service. In recent decades, the third-generation titanium alloys with nontoxic, low elastic modulus, and good plasticity have been introduced. They are new β-type titanium alloys because of the stable β-phase elements, such as molybdenum, thallium, niobium, and so on. Ti13Nb13Zr [1], Ti15Mo [2], Ti12Mo6Zr2Fe [3], Ti24Nb4Zr8Sn [4], Ti3Zr2Sn3Mo25Nb (TLM) [5], et al., belong to this type. TLM has a low elastic modulus to approximately 45 GPa. This value is close to that of bones (3~40 GPa) [6].

In space engineering and biomedical engineering, titanium and its alloys have a lot of applications. However, high friction coefficient, poor wear resistance, and low hardness are the restricting factors [7]. At present, shot peening is the standard finishing process, because shot peening improves the compressive residual stress and the hardness of surface [8]; however, the randomness and the inhomogeneity cause the rising of surface distortion and roughness. They are detrimental to the durability of materials. Thus, other surface modification methods have been developed recently.

Severe plastic deformation shall be induced when an external energy is applied on the surface of materials, such as ultrasonic, laser, and squeezing. The surface coarse grains are observed transforming into nanosized grains. This is called surface self-nanocrystallization (SSN). By means of mechanical methods, some of the SSN techniques have been hotly researched, such as surface mechanical attrition treatment [9], ultrasonic surface rolling processing [10], ultrasonic shot peening [11], laser shock peening [12], ultrasonic nanocrystalline surface modification [13], etc. The mechanical properties are generally improved by these treatments, including the hardness, tensile strength, residual stress, and so on [9,10,11,12,13]. The mechanism of grain refinement involves dislocation, twinning, and the new grain boundaries under high angle misorientation.

In fatigue tests, most failures are sensitive to the surface condition. Thus, it is essential to optimize the surfaces. The mechanisms of grain refinement have been revealed in the past [14,15]. It is accepted that the lattice structures and the stacking fault energy (SFE) decides the plastic deformation behavior and the dislocation in metals and alloys. The grain boundary serves as the importance of the plastic deformation of polycrystalline materials, especially in the fine grains with a size of micro- or nanometers [16]. It is recognized that the fatigue strength can be enhanced when the materials are subjected to these surface treatments [13,17]. The effects of stress ratio on the fatigue behavior of TC4 shows that the S–N curve presents a fatigue limit at the stress ratio of −1 and −0.5 [18]. The crack initiation can be simulated with the Murakami model, and the fatigue stress intensity factors are within the range of 6–8 MPa·m^1/2^. The estimated plastic zone at the crack tip is of a size similar to the primary α grains. The fatigue fracture behaviors of titanium alloys with fine grains in their surfaces are worth exploring.

In the present work, nanostructured surface layers were prepared by means of ultrasonic impact (UI) on TC4 and TLM specimens to study the effect of nanograins induced by plastic deformation on the fatigue behaviors of titanium alloys used in artificial bone. The metallographic phase, grain size, and some of the mechanical properties were investigated. The effects of ultrasonic impact on the rotating–bending fatigue were studied. Base on the theoretical constitutive relation, the mechanisms of the crack initiation of specimens subjected to UI treatment were analyzed.

## 2. Experimental Procedures

Two types of titanium alloys with different shape and content of body-centered cubic lattice β-phase were investigated. One is a typical α+β titanium alloy TC4, the other is a near β-type titanium alloy TLM. Table 1 gives the chemical composition of these two titanium alloys; the short dash means that the element is not contained. The heat treatments and the mechanical properties are listed in Table 2. All the specimens were furnace cooled at last.

Since the international patent of ultrasonic impact surface modification was authorized, it has been studied for nearly twenty years. The principle of this process uses a piezoelectric ceramics transducer (with a frequency of 20~30 kHz), and tens of thousands of strikes per unit area are applied to the surface. The surface impact generates severe plastic deformation and induces a thin nanograin layer in the surface. The ball tip is made of cobalt alloy with tungsten carbide (WC)coating and a size of 8 mm. The vibration amplitude is controlled as 30 μm. The vibration strike number is 48,000 times/mm^2^. Four groups of specimens, referred to as No-TC4, UI-TC4, No-TLM, and UI-TLM were prepared.

The cross-sections were observed by scanning electron microscopy (SEM, GeminiSEM 300, Zeiss, Hsu Koehn, Germany). The cross-sections were polished with waterproof abrasive paper until 2000 mesh. Then, they were fine ground with diamond powder. The specimens were etched in Kroll’s reagent (HNO_3_:HF:H_2_O = 3:6:90, vol.%) in the end. The severe plastic deformation layers were investigated with transmission electron microscopy (TEM) (Thermo Scientific, Talos F200i, Waltham, MA, USA). The TEM specimens were prepared by focused ion beam (FEI Helios Nanolab 600i, Columbus, OH, USA) underneath the surface. The surface hardness was tested by Vickers hardness tester (CMM-20E, Changfang, Chengdu, China). An atomic force microscope (AFM) (Bruker Dimension Icon, Billica, MA, USA) was used to picture the surface topography. X-ray diffraction (XRD) was used to examine the XRD patterns, using Rigaku X’pert pro MPD (Tokyo, Japan).

Rotating–bending fatigue test on the four groups of samples was conducted using an Ono fatigue test machine (Shimadzu, Fukuoka, Japan). Its frequency was 50 Hz. The fatigue test was operated at ambient temperature and with a stress ratio *r* of −1. The dimensions of the fatigue samples are given in Figure 1. The fracture surfaces were observed using scanning electron microscopy (JEOL JSM-6510LV & GeminiSEM 300, Zeiss, Hsu Koehn, Germany). Energy-dispersive X-ray (EDX) spectroscopy (Zeiss, Hsu Koehn, Germany) was used to analyze the crack initiation. According to the theories of bending, a stratification model and finite element method were applied to help explain the fish-eye cracks.

## 3. Results and Discussion

### 3.1. Observation of the Severe Plastic Deformation Layer

A certain depth of near-surface severe plastic deformation (SPD) layer is observed after the treatment of ultrasonic impact, normally. By means of TEM detection, it has been verified that nanocrystals can be obtained in this SPD layer [13,14,15]. The optical micrograph results of the microstructures are shown in Figure 2. After the solid solution treatment at 750 °C, β-type titanium alloy TLM mainly presents the equiaxed β-phase structures and secondary α-phase. The average size of β-phase grain in dark is about 20 μm. Ma et al. reported that the structures of TLM after solid solution or quenching are stable as other β-type titanium alloys [19]. To the two-phase (α + β) titanium alloy TC4, lamellar structure α-phase appears in large quantity after the solid-aging solution. The original size of it is approximate 100 μm. β-phase is intermingled among these lamellar structures. The mechanical properties of titanium alloy depend on the size, shape, and content of α-phase. Due to the β-phase, the plastic of TLM is better than TC4, but it has a lower strength. From the rheological trend and the deformed grains, it can be concluded that SPD layers with the depth of about 20 μm are achieved in the surface.

α-phase of titanium has a high stacking fault energy of more than 300 mJ/m^2^, β-phase with body centered cubic has 12 slip directions. Crystal–plane slip easily happens in most titanium alloys under the action of cyclic loads. Thus, the mechanism of forming nanograins in titanium alloy is mainly dislocation motion. Twinning is investigated in α-titanium sporadically [15]. TEM observations of the SPD layers induced by ultrasonic surface impact are shown in Figure 3. High density dislocations and dislocation cells are obviously investigated after the treatment of ultrasonic surface impact. The selected area electron diffraction (SAED) pattern of the area with a depth of 3 μm in the SPD layer is nearly concentric annulus. To both the two titanium alloys, there are thin dislocation walls. Figure 3c shows that the grains of TC4 are refined gradually, and the size reaches the nanometer scales. The lattice constants of SPD layers calculated from Figure 3d are much less than those which are given in reference [20]. The surface grains of TLM reach nanosize, similarly. Continuous slip bands are interweaved on the β-phase crystal. In general, the severe plastic deformation has helped the surface grains to transform into nanograins.

### 3.2. Surface Investigation

The surface features of these two titanium alloys subjected to ultrasonic impact are shown in Figure 4. From the 3D microscope images of a local region, it can be seen that the surface impact causes a concave–convex topography. It is reported that the width of the parallel lines induced by the process are accordant with the main feed of main shaft [21]. It appears as stick slip wear on TLM with a lower hardness. The surface roughness values of TC4 and TLM are 0.10 μm to 0.25 μm, separately. Comparing the original surface with the surface roughness of about 0.50 μm after polishing, the surfaces are smoothened. It is better than the results caused by an ultrasonic vibratory tip with a diameter of 2.38 mm, where the surface roughness of TC4 is even increased slightly [22]. This indicates that the effect of the impact tip’s size on the surface should be further analyzed.

Table 3 lists the surface hardness and surface residual stress of the two titanium alloys before and after the ultrasonic surface impact treatment. The values of both the hardness and residual stress are the average of six points of plane specimens subjected to the UI process. Normally, the hardness after UI treatment rapidly decreases to about a certain depth and then decreases gradually. The residual stress usually remains as a level in the SPD layer without any remarkable change, it disappears gradually in the grain refined layer and transforms to become weak tensile residual stress in deeper zones. Here, the surface hardness values of TC4 and TLM were improved by 14.7% and 23.5%, respectively. Compressive residual stress is an important factor for increasing the fatigue resistance. The position and shape of fish-eye crack initiation shall be influenced by the stress field and the original grains. The surface residual stress of TC4 and TLM are −508 MPa and −274 MPa, separately. Amanov, et al., reported that a compressive residual stress of more than 1000 MPa can be achieved by the UI process with static loads [23]. When the dislocation multiplication rate is balanced, the size of grains will not be reduced; the surface hardness shall reach their extreme value [24].

The XRD patterns of these two titanium alloys before and after ultrasonic surface impact are depicted in Figure 5. By means of this treatment, the intensity of characteristic peak α(101) and α(002) are obviously strengthened. From the pseudobinary phase diagram of β-type titanium alloy, it can be seen that α-phase shall exist in TLM. Due to the stable β-phase elements of Nb and Mo, the temperature of phase transition of β to α + β is about 710 °C. It is lower than that of TC4 (above 900 °C). Because both of them contain α- and β-phase, the main characteristic peaks are similar. This phenomenon is also investigated in other titanium alloys with nanograin surfaces [25]. In general, the peaks of β-phase are inconspicuous in contrast to those of α-phase. The phase transition is considered as another reason of the dramatic increase of strength and hardness, except in the grain refinement [26] (it is not visibly observed here). Taking the full width at half maximum of the selected characteristic peak, the average grain size can be estimated [27].

### 3.3. Fatigue Characteristics

The rotating–bending fatigue S–N curves of TC4 and TLM subjected to ultrasonic surface impact treatment are shown in Figure 6. Specimens that did not fracture are marked as run-outs. The cracks which initiate from the inside are marked with vertical bars. To both the two alloys, cracks of all the untreated specimens initiate from the surface, while they transform to become inner cracks at the long-life region of more than 10^6^ cycles. It is clear that ultrasonic surface impact enhanced the 10^7^ cycles fatigue strength of these two titanium alloys. The fatigue strength of TC4 is increased by 13.1%, and that of TLM is 23.7% improved. The fatigue limits are among the statistical section of (0.50~0.65)*σ*_b_ [5].

It is reported that all the cracks in rotating–bending fatigue initiate from the surface after ultrasonic nanocrystal surface modification [12]. By means of the laminated film method, the microcracks which are caused by ultrasonic surface impact processes are the reason for this phenomenon. The deformation overflow bands accelerate the surface crack’s propagation [22]. The energy field of crack depends on both the surface remodeling and the crystal slip in the severe deformation. Finer strike tip causes microcracks more easily because of the feed of knife with too-small steps [5]. With the better surface planeness, fish-eye cracks are common while the fatigue life is more than 10^6^ cycles. Figure 7, Figure 8 and Figure 9 are the SEM micrographs of the fatigue fracture surface of TC4 and TLM. Surface cracks usually initiate from the surface defects. The inner crack cores are at the zone with a depth of 100~250 μm. This area crosses the grain refined layer and the deformed coarse grain layer according to the regional division by Lu [14]. The compressed residual stress and hardness decrease rapidly in this area, where the plasticity and tenacity are weakened due to the severe plastic deformation in the UI process [22]. For TC4 subjected to UI treatment, there are high light white areas with an oblate arc shape appearing in the crack core, while the normal inner crack initiations are in a definite circular or elliptical rough area [18]. These white areas are extruded, deformed α-phases. For the near β titanium alloy TLM, inner crack cores usually form at the junction of crystals. In the crack initiation zone, slip bands are obviously observed in the deformed α-phases or nearby the triple grain boundary junction [28]. The fatigue stress intensity factor calculated with the Murakami theory is a little higher because the size of the facet area is large with a long narrow shape of ellipse. The development of fatigue cracks in the titanium alloy indicates that they usually ran along the grain boundary [29].

### 3.4. Mechanical Stratification Model Analysis

There are three analysis methods of the mechanical properties of nanostructures, including experimental method, molecular dynamics, and the modified theory of continuum mechanics. Due to the less calculation and the more accurate results, the modified theory of continuum mechanics is used to simulate the nanometer functional gradient materials widely. To both nonlocal theory and nonlocal strain gradient theory, the elastic modulus E is one of the important related items. It reflects the cohesion of atoms. With the studies of nanosized Fe, Cu, Ni, and Cu-Ni alloys, it is accepted that the elastic modulus of nanocrystalline metals are lower due to the relatively large volume fraction of pores [30]. Through the tensile test, it is found that the tensile strength of materials with nanograin surface shall be improved, while the elastic modulus has a tiny decrease [5,15]. The Poisson’s ratio is also less than that of polycrystalline materials [31]. The stress field of a model which is set with a gradient of elastic parameters is now discussed at the interface zone of the layers, to analyze the forming mechanism of inner cracks.

For example, there is a cantilever sandwich beam, as shown in Figure 10a; a vertical downward concentrated force is acted on the middle of its free end. Here, the surface of this beam has been treated by ultrasonic impact to nanograins. The elastic modulus of the inside base material and outside annulus are set as *E*_1_ and *E*_2_ (*E*_1_ > *E*_2_) and the Poisson’s ratios are defined as *μ*_1_ and *μ*_2_ (*μ*_1_ > *μ*_2_). Due to the circular cross-section, the centroid and the axis of symmetry are unchanged. The radius of curvature and the normal stress can still be calculated with Equation (1).
(1)$$\frac{1}{\rho }=\frac{M}{\sum E_{i}I_{i}},\;\sigma =\frac{E_{i}yM}{\sum E_{i}I_{i}}$$
where *M* is the bending moment, *σ* is the stress, *I_i_* is the inertia moment of the corresponding layer, *y* is the distance between the point to the centroid, and *E_i_* is the relative elastic modulus of the layer. The equivalent stiffness can be achieved by Equation (2). It is as the same as laminated plates.
(2)$$\sum E_{i}I_{i}=E_{1}I_{1}+E_{2}I_{2}$$

Thus, the normal stress at the separatrix *σ_sep_* can be calculated by Equation (3).
(3)$$\sigma_{sep}=\frac{E_{1}rM}{\sum E_{i}I_{i}}$$
where *r* is the radius of the inner circle. In a word, there shall be a saltation at the separatrix between the grain refined layer and the deformed coarse grain layer. This mechanical stratification model can be promoted to finer layer distinctions.

The stress analysis results via finite element method are given in Figure 10. It can be concluded that with decreasing the elastic modulus and Poisson’s ratio gradually, the maximum stress still appears at the top surface of the fixed end. It is in accordance with the typical theory of bending in the mechanics of materials. Noticeably, there is a turning point at the boundary, and a higher stress spans the separatrix in the gradient model. Combined with the drastic change of harness, residual stress, and the loss of plasticity in this zone, the inner cracks initiate easily. The core might be the severe plastic deformed α-phase or the slip intersections at the triple grain boundary junction. The crystal–plane slip on the body-centered cubic β-phase acts as the driving force of fish-eye cracks to both the two-phase (α + β) Ti alloys and the (near) β-type titanium alloys. Stress intensity factor of the crack tip is always used to study the crack propagation. The specimens with a dimensional specified notch can help to achieve the stress intensity factor [32,33]. The calculated value shows that it might be higher while the surfaces are transformed to be fine grains [22]. This also indicates that the crack initiations shall be more difficult in the specimens with nanograin surface.

## 4. Conclusions

After the treatment of ultrasonic surface impact (amplitude of 30 μm, strike number of 48,000 times/mm^2^), nanograins were achieved in the surfaces of both TC4 and TLM titanium alloys, mainly because of the dislocation motion. The mechanical properties were improved in many aspects, such as hardness, residual stress, and roughness. The rotating–bending fatigue strengths of TC4 and TLM subjected to ultrasonic impact were improved by 13.1% and 23.7%, separately. Fatigue cracks initiate from the surface of specimen before the fatigue life of 10^6^ cycles under high stress amplitude. By increasing the size of the ultrasonic impact tip, most of the inner cracks in the high life region present at the zone with a depth range of 100~250 μm. A long and narrow deformed α-phase usually appears in the inner crack core to TC4, while all the fish-eye cracks in TLM initiate at the triple grain boundary junctions. The change of normal stress in the gradient model helps in understanding these inner cracks. Slip deformation and the internal stress (besides the intersurface of grain refined layer and the deformed coarse grain layer) induce the cracks.

## Figures and Tables

**Figure 1 materials-14-00171-f001:**
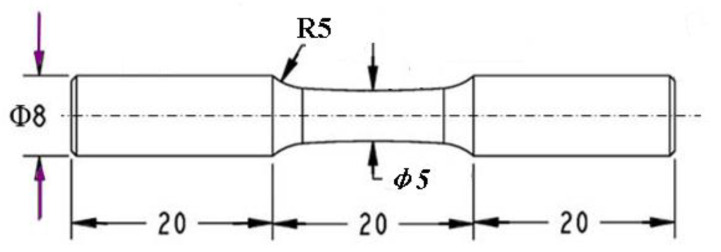
Dimension of the rotating–bending fatigue samples (in millimeter).

**Figure 2 materials-14-00171-f002:**
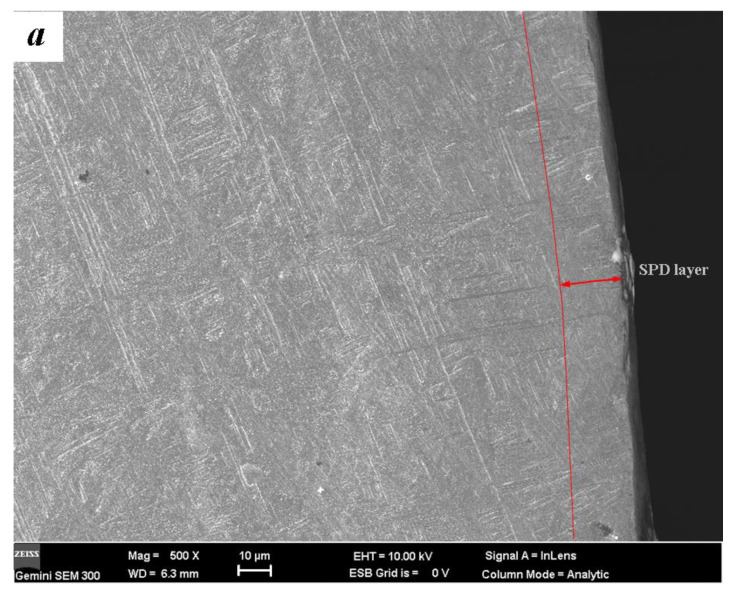
SEM images of metallographic observation: (**a**) Ti6Al4V; (**b**) Ti3Zr2Sn3Mo25Nb.

**Figure 3 materials-14-00171-f003:**
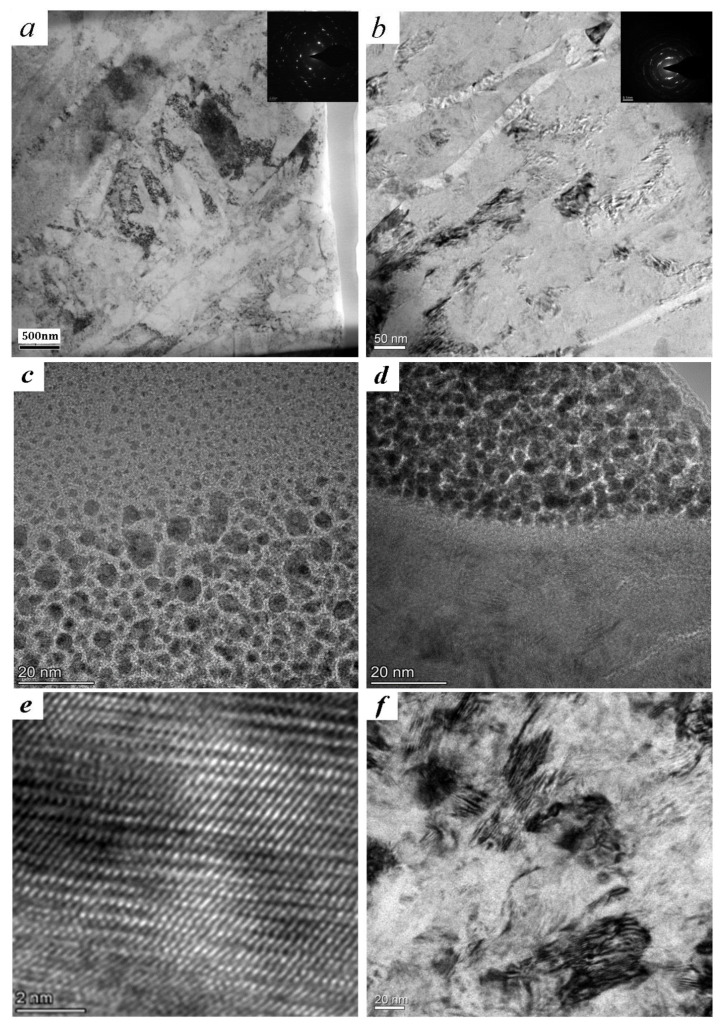
TEM observations: (**a**) image (depth of 3 μm) and diffraction pattern of TC4; (**b**) image (depth of 3 μm) and diffraction pattern of TLM; (**c**) image (depth of 1 μm) of TC4; (**d**) image (depth of 1 μm) of TLM; (**e**) the lattice image of Figure 3c; (**f**) the bright field image of TLM (depth of 4 μm).

**Figure 4 materials-14-00171-f004:**
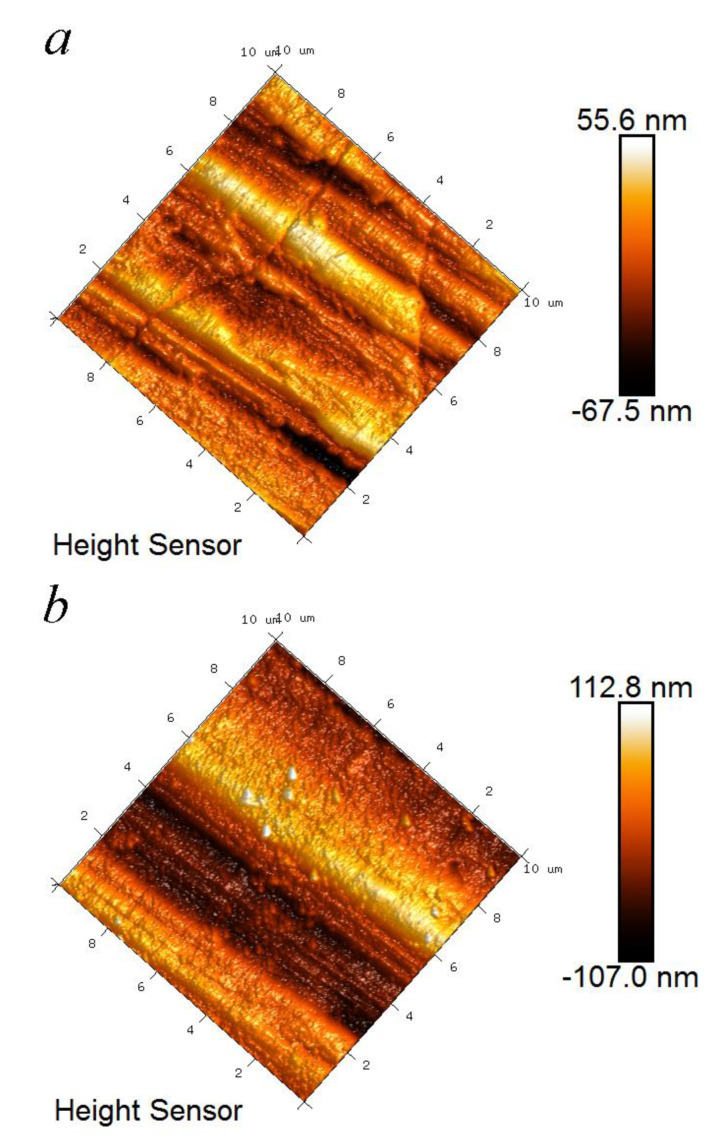
Surface topography by atomic force microscope (AFM) and surface roughness: (**a**) TC4; (**b**) TLM.

**Figure 5 materials-14-00171-f005:**
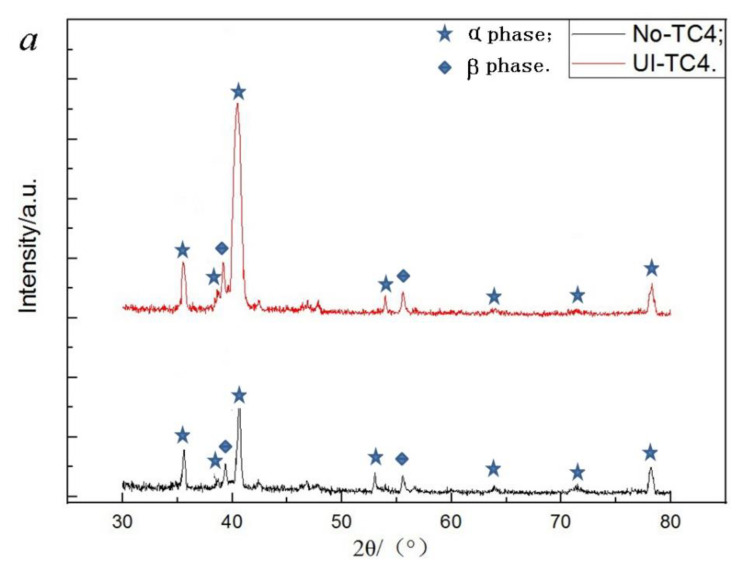
Analysis of XRD patterns: (**a**) TC4; (**b**) TLM; (**c**) pseudobinary phase diagram of β-type titanium alloy.

**Figure 6 materials-14-00171-f006:**
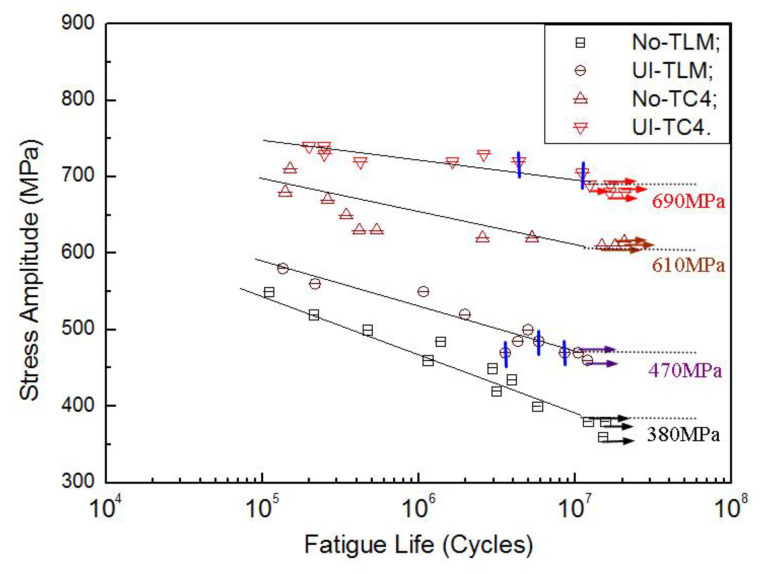
Rotating–bending fatigue S–N curves of TC4 and TLM titanium alloys.

**Figure 7 materials-14-00171-f007:**
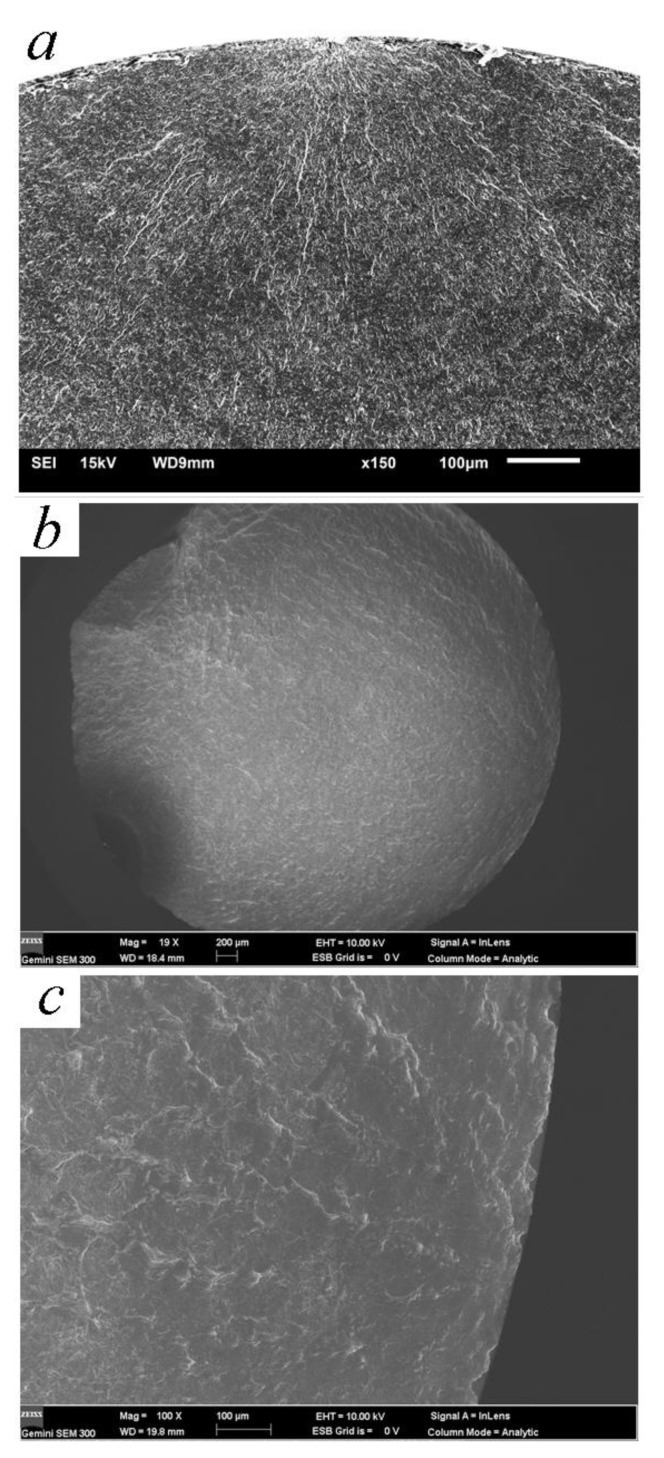
Surface cracks of TC4 and TLM: (**a**) TC4-ultrasonic impact (UI), 730 MPa, 2.49 × 10^5^ cycles; (**b**) TLM-UI, 550 MPa, 1.07 × 10^6^ cycles; (**c**) magnification of the crack initiation of Figure 7b.

**Figure 8 materials-14-00171-f008:**
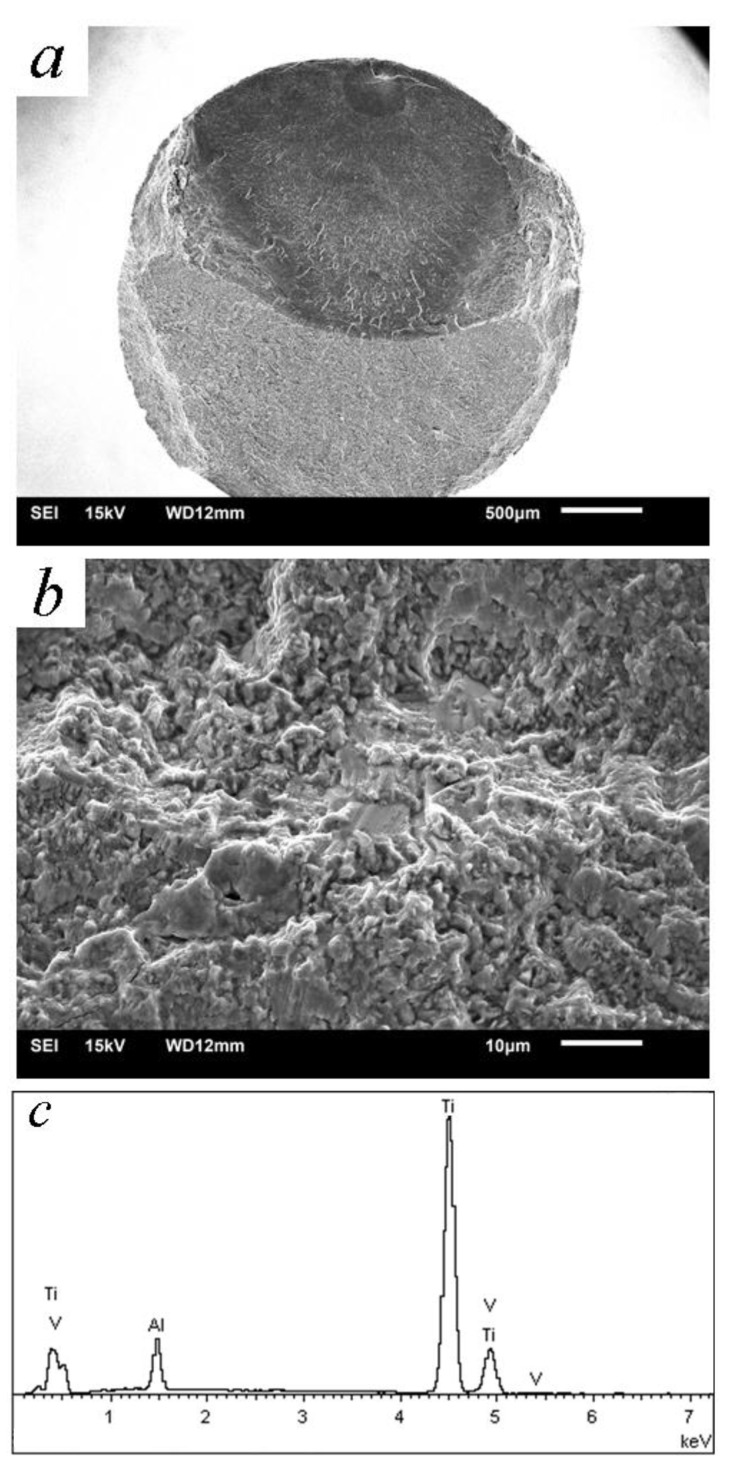
Fracture surface of TC4 subjected to UI treatment (720 MPa, 4.36 × 10^6^ cycles): (**a**) overall view of the fracture surface; (**b**) crack initiation with facets; (**c**) EDX analysis of crack core.

**Figure 9 materials-14-00171-f009:**
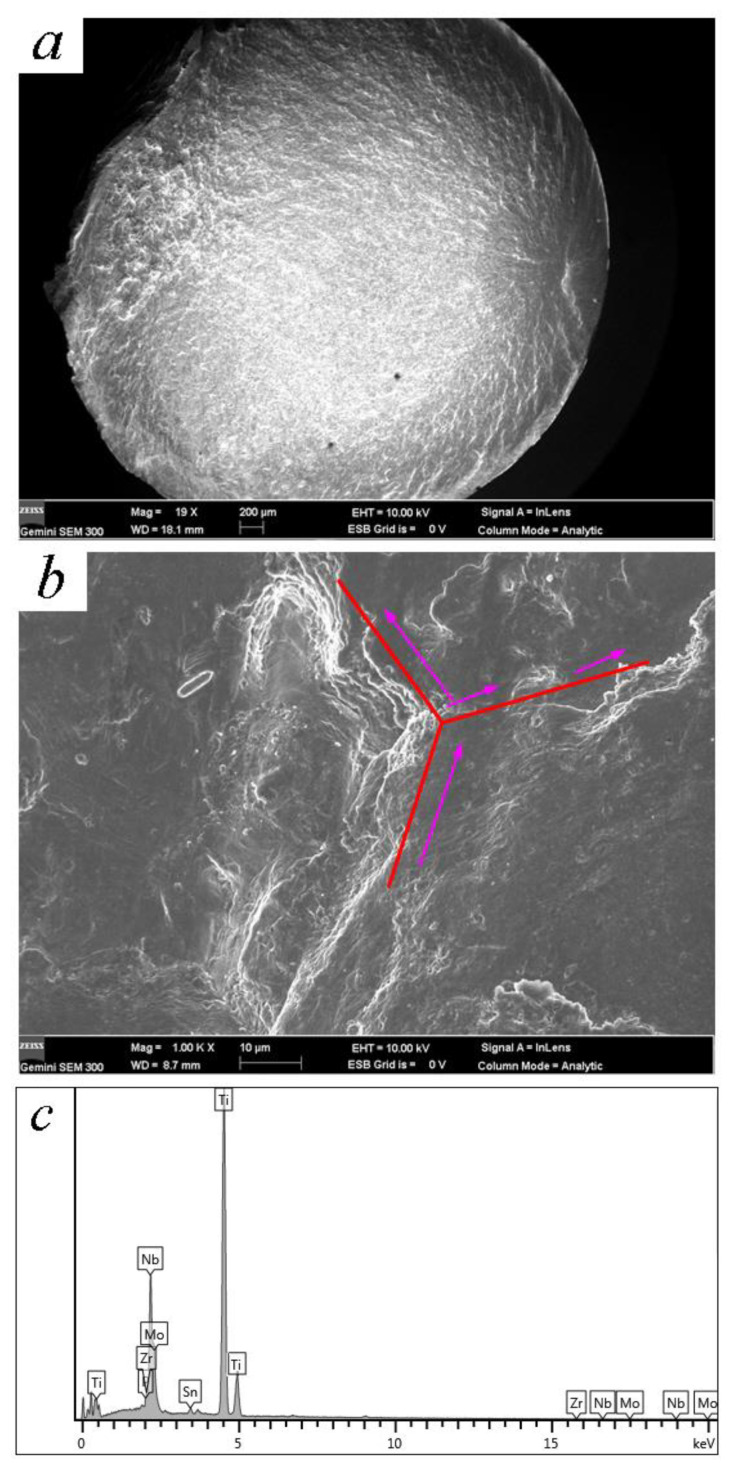
Fracture surface of TLM subjected to UI treatment (485 MPa, 5.83 × 10^6^ cycles): (**a**) overall view of fracture surface; (**b**) illustration of the crack initiation; (**c**) EDX analysis of crack core.

**Figure 10 materials-14-00171-f010:**
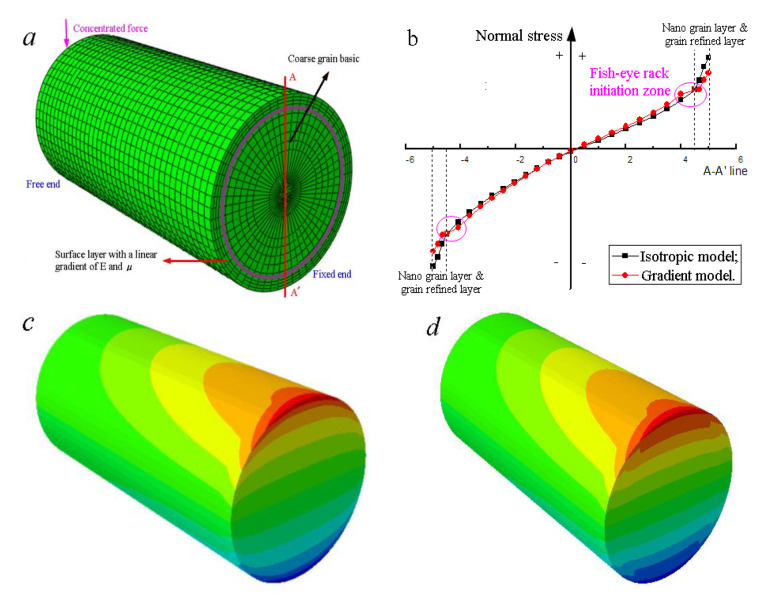
Simulation analysis of bending normal stress: (**a**) stratification model of a cantilever beam; (**b**) maximum normal stress is reduced with surface nanograins; (**c**) distribution of normal stress of isotropic model (E and μ of surface are a little lower than that of the basic); (**d**) distribution of normal stress of gradient model (both E and μ of surface are with gradient changes).

**Table 1 materials-14-00171-t001:** The chemical properties (mass %) of Ti6Al4V (TC4) and Ti3Zr2Sn3Mo25Nb (TLM).

Material	C	O	N	H	Fe	Mo	V	Al	Zr	Sn	Nb	Ti
TC4	0.006	0.192	0.008	0.001	0.02	—	4.404	6.409	—	—	—	rest
TLM	0.015	0.16	0.007	0.003	—	3.10	—	—	3.07	2.09	24.8	rest

**Table 2 materials-14-00171-t002:** The heat treatment and mechanical properties of TC4 and TLM.

Material	Heat Treatment	σ_0.2_/MPa	σ_b_/MPa	δ/%	ψ/%
TC4	980 °C/1 h + 650 °C/4 h	850	925	16	26
TLM	750 °C/30 min + 510 °C/4 h	567	721	19.5	71

**Table 3 materials-14-00171-t003:** Surface hardness and residual stress of TC4 and TLM.

Material	Hardness (HV, 50 g, 20 s)		Residual Stress (MPa)	
	Before UI	After UI	After Polishing	After UI
TC4	340	390	−35	−508
TLM	238	294	−12	−274

## Data Availability

We confirm that all the data are available.

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
