# Peer review of "Influence of Fine Grains on the Bending Fatigue Behavior of Two Implant Titanium Alloys"

_materials, 2020, doi:10.3390/ma14010171_

Round 1
Reviewer 1 Report
- Please explain all abbreviations and markings at work, e.g. SPD, SAED, UI, and others.
- Fig. 1 - are the dimensions in mm? Add this information in your drawing caption.
- On which device was the hardness tested, at what load, etc.?
- Please provide information on which equipment the fatigue tests were performed and additional information about the tests carried out.
- What were the dominant plastic or brittle cracks?
- Fig. 10c, d - please provide approximate stress values that correspond to the colors on the presented models.
- What software was used for the FEM calculations and please expand the information on the calculations. What material model was used, number of finite elements, etc.?
- It would also be worthwhile to quote the following papers: 1) Małecka J., Rozumek D.: Metallographic and mechanical research of the O-Ti2AlNb alloy. Materials Vol. 13, 3006, 2020, 2) Rozumek D., Faszynka S., Surface cracks growth in aluminum alloy AW-2017A-T4 under combined loadings. Engineering Fracture Mechanics 2020, 226, 3) ASTM E 739-80, Standard practice for statistical analysis of linearized stress-life (S-N) and strain-life (E-N) fatigue data, in: Annual Book of ASTM Standards, Vol.03.01. Philadelphia, 1989, pp. 667-673.
Author Response
To reviewer 1:
Thank you very much for the friendly comments. Hope that everything is all right during the outbreak of COVID-2019. Enjoy the Christmas Holiday.
The pointed weaknesses are corrected and explained.
Best wishes,

Reviewer 2 Report
The article addresses interesting and up-to-date research topis that may have wide area of usage in medical-implants industry. Generally, the article is well writen and the idea is clearly presented. I only have some minor suggestions for improving the article:
1.) Page8, line 176: the xx should be replaced by the authors Amanov and Umarov.
2.) Page 10, Figure 6: there is significant scatter linked to the S-N data. The authors may estimate it by using the Weibull's PDF. Fore example, see reference: Jernej Klemenc, Matija Fajdiga: Estimating S–N curves and their scatter using a differential ant-stigmergy algorithm. International Journal of Fatigue 43 (2012) 90–97.
3.) Page 10, line 207: typografic error: "chich" => "which".
4.) Page 14, line 239: A few word on motivation for performing the FE analysis should be given here (one sentence).
5.) Page 16, lines 273-281: It would be nice if the discussion of the FE results is also linked to Figure 8a.
Author Response
To reviewer 3:
Thank you very much for the friendly comments. Hope that everything is all right during the outbreak of COVID-2019. Enjoy the comming Christmas holiday.
The weaknesses are corrected.
Best regards,

Round 2
Reviewer 1 Report
The Authors took into account all comments and suggestions of the Reviewer. Please also add comments (remarks) to the work, not only in responses to the reviewer! I recommend publishing the article in the Materials.
Author Response
To Reviewer 1,
Thank you for the comments. The manuscript has been revised as the listed several points in the attachment.
Best wishes,
